# The Effectiveness of Compassion Focused Therapy for the Three Flows of Compassion, Self-Criticism, and Shame in Clinical Populations: A Systematic Review

**DOI:** 10.3390/bs15081031

**Published:** 2025-07-29

**Authors:** Naomi Brown, Katie Ashcroft

**Affiliations:** 1Berkshire Healthcare Foundation Trust, Bracknell RG12 7FR, UK; 2Department of Psychology, Royal Holloway University of London, Egham TW20 0EX, UK; katie.ashcroft@rhul.ac.uk

**Keywords:** clinical populations, compassion-focused therapy, compassion flows, self-compassion, self-criticism, shame, systematic review

## Abstract

Compassion Focused therapy (CFT) is designed to reduce shame (internal and external) and self-criticism while enhancing the three flows of compassion (compassion to others, from others, and for the self). This systematic review evaluated the effectiveness of CFT on these core theoretical constructs in adult clinical populations. A systematic search of three databases (2000–2024) identified 21 studies (*N* = 450) meeting the inclusion criteria. The studies were narratively synthesised, and quality was assessed using the EPHPP tool. Consistent improvements in self-compassion (*g* = 0.23–4.14) and reductions in self-criticism (*g* = 0.29–1.56) were reported. Reductions in external shame were also observed (*g* = 0.54–1.22), though this outcome was examined in fewer studies. Limited and inconsistent evidence was found for internal shame and interpersonal compassion flows (compassion to and from others), with only a small number of low- to moderate-quality studies addressing these outcomes. Follow-up effects were rarely assessed, and comparator groups were limited. Most interventions were group-based and of variable methodological quality, with frequent selection bias, small sample sizes, and limited demographic diversity. Overall, CFT shows promise for targeting self-directed processes in clinical populations, though stronger evidence is needed to understand its effects on relational components of compassion. Future research should adopt standardised measures, improve methodological rigour, and recruit more diverse samples.

## 1. Introduction

Compassion Focused therapy (CFT), developed by [24] ([24]), is a psychological intervention that addresses mental health difficulties through compassion cultivation. Drawing from evolutionary psychology and neuroscience, CFT targets common factors in mental health conditions, particularly self-criticism and shame, through fostering compassionate behaviour ([35]; [29], [32]).

Within CFT, self-criticism and shame are understood as barriers to wellbeing ([39]). Self-criticism involves harsh self-judgment, while shame encompasses feelings of unworthiness. Shame can be directed internally (shame directed towards oneself) or externally (perceiving others as shaming; [11]; [29]; [58]). CFT theory suggests that both internal and external shame activate the threat system, creating self-protective responses that paradoxically maintain psychological distress ([40]). CFT interventions theoretically engage with internal shame through developing self-compassion and a kinder internal relationship, while addressing external shame by fostering secure connections and receptivity to compassion from others ([40]). Persistent self-judgment is viewed as a barrier to self-compassion and emotional regulation, with self-criticism and shame directly contributing to and maintaining psychological distress ([39]; [40]).

A CFT approach defines compassion as sensitivity and commitment to alleviating suffering ([29]). CFT theory proposes three emotion regulation systems: threat (safety-seeking), drive (resource-seeking), and soothing (affiliation-seeking). The model suggests that cultivating a compassionate motivation, balances these systems by activating the soothing system ([40]). The soothing system is then used counteract excessive threat activation and behaviour motivated by threat ([29], [32]).

Compassion in CFT operates through three distinct ‘flows’, each serving vital yet complementary functions ([29]). These include (a) Compassion for Others: nurturing social connections through caring behaviour directed toward others; (b) Compassion from Others: developing receptivity to care and support from others, which affirms one’s worthiness of compassion; and (c) Self-Compassion: cultivating emotional resilience through treating oneself with kindness and understanding. These flows work together to promote psychological wellbeing by addressing both interpersonal and intrapersonal dimensions of compassion ([32]).

### 1.1. Previous Reviews and Current Gaps

Early systematic review evidence ([56]) demonstrated CFT’s efficacy for reducing self-criticism and improving self-compassion in clinical populations. However, this review excluded key theoretical mechanisms like shame and interpersonal compassion flows. Later reviews ([50]) examined multiple ‘compassion-based’ interventions together, limiting conclusions about CFT’s unique therapeutic mechanisms. This trend of grouping compassion interventions persists in recent reviews ([85]; [89]), despite concerns that this approach obscures the unique mechanisms of each intervention ([49]). Recent CFT-specific reviews in clinical populations ([15]; [64]) have focused on general symptoms rather than CFT’s theoretical mechanisms of change, such as compassion flows, self-criticism, and shame ([40]). This gap limits our understanding of how CFT achieves its therapeutic effects. The current review addresses these gaps by synthesising evidence for CFT’s impact on its core theoretical constructs in clinical populations. Given the limited RCT evidence for shame and compassion flows, the review included both RCTs and other study designs to provide a comprehensive evaluation.

### 1.2. Aims

This systematic review evaluates the effectiveness of CFT for clinical populations, focusing on three key domains: the three flows of compassion, self-criticism, and shame.

Primary research questions examine CFT’s effectiveness in changing
The three flows of compassion (self-compassion and interpersonal compassion: compassion from others, and compassion to others);Self-criticism;Shame (internal and external).

The review examines changes in these outcomes from baseline to post-intervention and follow-up, analysing published and unpublished CFT studies from 2000 to 2024 through narrative synthesis.

## 2. Materials and Methods

This systematic review followed the PRISMA guidelines ([69]) and was pre-registered with PROSPERO (CRD42023441154).

### 2.1. Eligibility Criteria

Studies were selected using the PICOS framework (see Table 1). Clinical populations were identified through diagnostic evidence, clinical settings, or appropriate screening measures. Studies were included if they provided diagnostic evidence or were conducted in a clinical setting, even if screening measures were absent. Seven papers were determined to be from a clinical population based on diagnostic evidence or setting. Key inclusion criteria were as follows:-Adult clinical populations (≥18 years)-CFT interventions following Gilbert’s model-Quantitative measures of compassion flows, self-criticism, or shame-Any study design except single case studies-Published and unpublished studies to reduce publication bias

Studies needed to be available in English for inclusion due to practical limitations in translation resources and to ensure accurate data extraction and quality assessment.

### 2.2. Search Strategy

A comprehensive search strategy was developed using the PICOS framework ([63]) in consultation with a research librarian. The searches covered three major databases (PubMed, PsycInfo EBSCOhost, and Web of Science) from 2000 to 2024, using carefully selected terms for CFT interventions and outcomes of interest. A backward citation search was also conducted. Databases were searched in August 2023, February 2024, and October 2024. Search terms combined “Compassion focused therapy” variants with outcome-related terms (e.g., shame, self-criticism, and compassion flows). The complete search methodology, including all terms, Boolean operators, and database-specific adaptations, is provided in Appendix A.

While the protocol indicated inclusion of unpublished dissertations and theses to reduce publication bias, the search was limited to three major academic databases (PubMed, PsycInfo EBSCOhost, Web of Science). Although PsycInfo EBSCOhost and Web of Science include selected unpublished dissertations, comprehensive grey literature searching would require additional specialised databases such as ProQuest Dissertations & Theses Global, which was beyond the scope of this review. All 21 studies included in the final synthesis were peer-reviewed published articles.

### 2.3. Selection Process

The study selection followed a systematic, multi-stage process to identify relevant CFT intervention research. Initial database searches of PubMed, PsycInfo EBSCOhost, and Web of Science identified 1209 records, supplemented by 1208 records from backward citation searching. After removing duplicates, 1724 unique records underwent title and abstract screening against eligibility criteria.

Full-text assessment of 66 potentially eligible reports led to 21 studies meeting the inclusion criteria. Primary reasons for exclusion included non-clinical populations (*n* = 26), incorrect outcomes (*n* = 13), and inappropriate interventions (*n* = 1). Five papers could not be retrieved for full-text review despite attempts to contact the authors, and these were excluded from the analysis. Author contact clarified the eligibility of three studies. All 21 studies included in the final synthesis underwent a complete full-text review.

To ensure selection reliability, an independent reviewer (a trainee clinical psychologist) screened 10% of the database records (*n* = 66), selected through random number generation. This process achieved complete agreement (*κ =* 1.00), supporting the robustness of the screening procedure.

A PRISMA flow diagram detailing the selection process and search strategy is provided in Figure 1.

### 2.4. Data Extraction

Data extraction covered publication details, study characteristics, sample demographics, intervention details, study design, outcome measures, and outcomes. The studies were assessed for effectiveness using statistical significance and standardised mean differences at post-intervention and follow-up. All data were manually extracted and tabulated.

One study did not report participant gender data ([1]). Despite author contact, this information remained unavailable; however, the study was retained as it met the inclusion criteria and included a three-month follow-up assessment, which provided valuable longitudinal data for the synthesis.

Outcomes were categorised into three domains: compassion flows (self-compassion, compassion from others, and compassion to others), self-criticism (including inadequate self, hated self, and self-reassurance), and shame (internal and external).

Effect sizes were computed by the authors for all studies using available statistics (means, standard deviations, sample sizes, test statistics) to ensure precision and comparability across studies. Effect sizes were calculated using reported statistics following established methods ([7]). Authors were contacted for missing data when necessary.

Effect sizes were calculated using Cohen’s *d* for parametric tests and Rosenthal’s *r* for non-parametric tests, with Hedges’ *g* correction for small samples ([13]; [76]; [44]; [45]). Standard interpretations were used for effect size magnitude (very small, small, medium, large), as outlined in Table 2.

### 2.5. Quality Assessment

Study quality was evaluated using the Quality Assessment Tool for Quantitative Studies developed by the Effective Public Health Practice Project (EPHPP; [82]), which assesses selection bias, study design, confounders, blinding, data collection methods, attrition, intervention integrity, and analyses. Intervention integrity and analyses do not contribute to the global study quality rating. Global ratings were assigned as strong (low risk), moderate (some concerns), or weak (high risk). The EPHPP tool uses the term ‘blinding’; however, this paper uses the term ‘masking’ for inclusivity. Both terms refer to procedures that keep assessors and/or participants unaware of group allocation or study hypotheses to reduce bias. Due to ethical constraints in psychological intervention research, the overall grading and grading excluding ‘masking’ were reported to offer a more equitable assessment.

An independent and anonymous reviewer (a trainee clinical psychologist) screened 20% of the 21 included reports (*n* = 4) against the quality assessment tool to ensure inter-rater reliability. During the first discussion, a complete agreement was reached (100%; κ = 1.00).

### 2.6. Synthesis Methods

Due to heterogeneity in study designs and outcomes, a narrative synthesis approach was adopted. Results were organised by outcome domain (compassion flows, self-criticism, and shame) and synthesised using descriptive statistics and effect sizes. Hedges’ *g* effect sizes were summarised visually using a forest-style plot. The forest plot illustrates the direction and magnitude of effects across studies but does not reflect a formal meta-analytic synthesis. Only studies in which Hedges’ *g* could be computed were included in the forest plot to ensure consistency and interpretability of the effect size estimates. Studies without sufficient data to compute Hedges’ *g* could not be visualised in this format and were therefore excluded from the figure. Study quality ratings were integrated into the synthesis to evaluate the certainty of evidence.

## 3. Results

### 3.1. Study Characteristics

The 21 included studies were conducted in nine countries: Australia (*n* = 1), Colombia (*n* = 1), Denmark (*n* = 1), Iran (*n* = 4), Italy (*n* = 1), Japan (*n* = 1), the Netherlands (*n* = 1), the UK (*n* = 9), and the USA (*n* = 2) from 2006 to 2024. Sample sizes ranged from 7 ([12]) to 91 participants ([22]). Of the 450 participants, 72.88% identified as women (*n* = 328), aged 18 to 89. One study did not report gender ([1]).

The studies included eight adult clinical populations: depression (*n* = 4), personality disorder (*n* = 3), bipolar disorder (*n* = 2), post-traumatic stress disorder (PTSD; *n* = 1), anxiety (*n* = 1), schizophrenia/psychosis (*n* = 1), obsessive-compulsive disorder (OCD; *n* = 1), and prolonged grief disorder (*n* = 1). Seven studies included transdiagnostic samples. The study settings included community (*n* = 7), specialist services (*n* = 6), secondary mental health (*n* = 5), and inpatient facilities (*n* = 3).

Most studies were cohort designs (*n* = 12; [1]; [12]; [22]; [33]; [39]; [43]; [48]; [53]; [60]; [61]; [62]; [65]), with five RCTs ([4]; [8]; [23]; [47]; [78]), two multiple baseline designs ([70]; [71]), one case-controlled clinical trial with a waitlist control ([67]), and one cohort analytic study with treatment-as-usual as the control group ([81]).

Thirteen studies (61.90%) included follow-up assessments ranging from one month to one year. Twenty studies reported retention rates averaging 77.18% (range: 24.39–100%). The study characteristics are presented in Table 3, and the study designs and assessment time points are detailed in Table 4.

### 3.2. Intervention Characteristics

All interventions followed Gilbert’s CFT model ([25], [28]; [35]; [37]; [39]; [38]). Eight studies adapted protocols for specific populations ([12]; [23]; [33]; [47]; [53]; [60]; [67]; [70]).

Group therapy was predominant (*n* = 19), averaging 12.42 weekly sessions (range = 5–25), each lasting 1–2.5 h. Two studies used individual formats ([8]; [23]). Qualified clinical psychologists delivered most interventions (*n* = 14), often with co-facilitators. Thirteen studies reported facilitator CFT training, with ten receiving ongoing supervision from Paul Gilbert ([4]; [12]; [22]; [33]; [39]; [47]; [48]; [60]; [61]; [70]).

### 3.3. Quality Appraisal

#### 3.3.1. Risk of Bias Within Studies

Initial quality assessment rated all studies as weak due to unavoidable participant awareness of intervention (the masking domain). Excluding masking, 14 studies were upgraded to moderate quality (see Figure 2; [8]; [12]; [22]; [23]; [33]; [39]; [47]; [48]; [53]; [60]; [61]; [67]; [70]; [78]). Selection bias and confounders were consistently rated as weak across all studies. Six studies reported intervention consistency measures ([4]; [8]; [22]; [23]; [61]; [78]). Three indicated a high likelihood of co-intervention, as participants received CFT alongside another intervention ([22]; [1]; [78]).

#### 3.3.2. Risk of Bias Across Studies

Analysis across studies revealed consistent patterns in quality ratings. While confounders, data collection methods, withdrawals, and dropouts showed strong ratings, selection bias and masking were consistently rated as weak across all studies (see Figure 3). However, the study designs received a mostly moderate rating, indicating some bias concerns. This was due to recruitment methods and the inability to mask participants from psychological interventions. Reporting completeness varied, with several studies requiring author contact for additional methodological details.

### 3.4. Results Overview

Table 3 provides an overview of the study designs, outcome measures, assessment time points, and main findings for each included study. Effect sizes were computed for each study using standardised methods to maximise precision and ensure comparability. A forest-style summary of the standardised effect sizes (Hedges’ *g*) is presented in Figure 4. In Figure 4, outcomes are grouped by construct, illustrating the magnitude and variability of CFT effects across compassion, self-criticism, and shame.

#### 3.4.1. Primary Outcomes

##### Compassion Flows

Measurement and Evidence Quality. Sixteen studies measured at least one compassion flow. Self-compassion was primarily assessed using validated measures: the Self-Compassion Scale (SCS; [66]), its short form (SCS-SF; [74]), and the Compassionate Engagement and Action Scales (CEAS; [34]). Some studies employed non-standardised measures like interval contingent diaries and visual analogue scales ([39]; [48]; [81]). Study quality varied, with six moderate-quality and four weak-quality studies contributing to the self-compassion evidence base. The remaining compassion flows were measured less frequently (compassion from others: *n* = 3, compassion to others: *n* = 4), using either the CEAS or visual analogue scales, with studies ranging from weak to moderate quality ([4]; [22]; [33]; [81]). The underrepresentation of compassion to and from others reflects the study designs of the included research. The researchers predominantly selected self-compassion measures over available interpersonal compassion scales.

Effectiveness Results. Self-compassion showed the strongest evidence among compassion flows. Ten studies demonstrated significant improvements (*g* = 0.23–4.14; [1]; [4]; [22]; [23]; [39]; [48]; [61]; [62]; [78]; [81]). Four controlled studies showed superiority to waitlist or treatment-as-usual conditions ([4]; [23]; [78]; [81]). Follow-up data from four studies indicated maintenance of gains in most cases ([23]; [61]; [70]; [71]).

Evidence for the remaining compassion flows was more limited in both quantity and quality. Three studies examining the ability to receive compassion from others found small improvements (*g* = 0.26–0.36; [4]; [22]; [33]), with moderate-quality studies suggesting some reliability in these findings. However, the results for compassion to others were less convincing, with only two of four studies reporting very small positive effects (*d* = 0.14–0.18; [4]; [81]), and these came from methodologically weaker studies. This pattern of limited measurement, weaker methodology, and inconsistent effects prevents firm conclusions about CFT’s impact on all compassion flows.

##### Self-Criticism

Measurement and Evidence Quality. Fifteen studies assessed self-criticism, primarily using the Forms of Self-Criticism/Self-Reassuring Scale (FSCRS; [36]). The FSCRS assesses self-criticism across three subscales: ‘inadequate self’, ‘hated self’, and ‘reassured self’. Additional measures included the Level of Self-Criticism Scale (LOSC; [83]) and the Depressive Experiences Questionnaire (DEQ; [77]). Most studies (*n* = 9) achieved moderate-quality ratings despite selection bias limitations ([22]; [23]; [33]; [39]; [47]; [48]; [60]; [61]; [70]). As such, there are some concerns about bias across self-criticism outcomes.

Effectiveness Results. Evidence for improvements in self-criticism was consistent across measures and studies. Of the eleven studies using the FSCRS, five reported significant reductions in ‘inadequate self’ (*g* = 0.29–1.56; [22]; [48]; [61]; [62]; [70]), six showed decreased ‘hated self’ (*g* = 0.33–0.97; [22]; [39]; [48]; [60]; [61]; [62]), and six demonstrated increased ‘reassured self’ (*g* = 0.40–1.18; [22]; [39]; [48]; [60]; [61]; [70]). Controlled studies from two studies somewhat strengthened these findings. Some of the subscales showed superiority to waitlist and treatment-as-usual conditions ([47]; [78]). Follow-up data from two studies showed mixed sustainability. One study reported large improvements at follow-up (*g* = 0.92–1.70; McLean et al., 2022), and another reported diminished effects ([70]). A significant decrease in self-criticism was also found post-intervention in one study using the DEQ ([22]). Out of the two studies using the LOSC, only one found significant reductions compared to a waitlist control; however, the results were not maintained at follow-up ([23]). The improvements demonstrated across the studies suggest that CFT may reduce self-criticism in the short term; however, the moderate study quality indicates that results should be interpreted with caution.

##### Shame

Measurement and Evidence Quality. Seven studies examined shame outcomes, with most achieving moderate-quality ratings ([39]; [48]; [53]; [60]; [61]; [62]). External shame was assessed in six studies using the Other as Shamer Scale (OAS; [42]). Internal shame was measured in one study ([48]) using the Internalised Shame Scale (ISS; [14]). Global shame was measured in one study ([61]) using the External and Internal Shame Scale (EISS; [19]). As with the interpersonal compassion flows, internal shame was underrepresented in the included studies. The researchers typically selected external shame measures over tools assessing multiple dimensions of shame, such as the EISS ([19]). Even when the EISS was used ([61]), the available subscale data were not analysed. The predominance of moderate-quality studies suggests that most shame outcomes include some concerns with bias.

Effectiveness Results. External shame showed the most consistent evidence, with significant reductions across all six studies (*g* = 0.54–1.22; [39]; [48]; [53]; [60]; [61]; [62]). Five of these studies were of moderate quality, suggesting some concerns about bias and reduced confidence in these outcomes. Follow-up data from two studies showed either maintained ([61]) or enhanced reductions over time ([60]). The single moderate-quality study measuring internal shame reported large reductions (*g* = 1.28; Judge et al., 2012). Global shame showed similarly large, sustained improvements (*g* = 1.32–1.67) in the one study measuring this construct ([61]). The limited measurement of these shame types prevents broader conclusions about CFT’s effectiveness beyond external shame.

## 4. Discussion

This systematic review assessed the effectiveness of CFT interventions on the three flows of compassion, self-criticism, and shame in clinical populations. Across the 21 studies included, significant improvements in self-compassion were reported, with most studies showing medium to large effect sizes. Consistent reductions in self-criticism were also observed, particularly feelings of inadequacy and self-hatred,. External shame decreased significantly in studies measuring this outcome, though internal shame was assessed less frequently. Evidence regarding compassion to others and receiving compassion from others was limited and mixed. Some studies indicated maintained or increased effects at follow-up; however, few included comparator groups. Where comparisons were available, results suggested potentially greater improvements in self-compassion relative to waitlist and treatment-as-usual controls. By examining CFT interventions in relation to their core theoretical constructs, these findings address gaps in previous reviews that grouped a range of compassion-based interventions or focused solely on general symptom reduction.

### 4.1. Primary Findings

The review identified outcome patterns that partially support CFT’s theoretical framework. Self-compassion and self-criticism emerged as the most reliably supported outcomes, likely reflecting their frequent inclusion across studies. The findings were relatively convergent across studies, though methodological limitations were common.

Self-compassion had the most comprehensive evidence base. Seventeen studies reported significant improvements, with effect sizes ranging from small to large (*g* = 0.23 to 4.14). Although most studies were rated as weak to moderate quality due to small samples, a high risk of bias, and limited demographic diversity, the consistency of findings increases confidence in this outcome. These results align with prior reviews highlighting CFT’s efficacy in enhancing self-compassion ([50]; [56]). The findings support [29]’s ([29]) theory that self-compassion activates the soothing system, counteracting threat-based responses. This consistency likely reflects self-compassion’s central role in CFT protocols and the use of validated measures. However, most studies (*n* = 11) used [66]’s ([66]) Self-Compassion Scale [66]’s ([66]) scale focuses on self-directed compassion. Therefore, studies using this scale to measure self-compassion potentially overlook CFT’s broader theoretical scope. The limited demographic diversity further restricts generalisability, with 72.88% of participants identifying as female across the nine countries represented. Overall, self-compassion is the most well-supported outcome in the current evidence base.

Self-criticism also showed consistent improvement across 14 studies, with effect sizes ranging from small to large (*g* = 0.29 to 1.56) across subscales, including inadequate self, hated self, and reassured self. These studies were typically of moderate quality, though common issues included small sample sizes and risk of bias. Despite this, the consistency supports CFT’s effectiveness in reducing self-critical tendencies, echoing previous reviews ([56]; [64]). These results align with theoretical claims that compassion practices reduce self-criticism by encouraging more balanced self-evaluation ([40]). However, the use of varied measures and subscales complicates direct comparisons. The gender imbalance noted for self-compassion also applies here. This limitation intersects with other methodological concerns such as selection bias, high attrition, and lack of masking ([15]; [64]). Although some evidence suggests gender differences in experiences of self-criticism ([75]; [84]), few studies have explored gender-based variation in CFT outcomes. Future research should investigate how people of different genders experience and respond to CFT to ensure that findings are relevant across diverse populations.

In contrast to the stronger findings for self-compassion and self-criticism, external shame was assessed in only six studies. These reported significant reductions, with medium to large effect sizes (*g* = 0.54 to 1.28). However, the studies were small and of moderate quality. While the results suggest that compassion may reduce shame, the mechanisms—such as increased social safeness or reduced fear of negative evaluation—were not directly examined ([32]). The limited number and quality of studies reduce confidence in this outcome compared to others. Demographic homogeneity further limits generalisability, possibly obscuring cultural differences in the experience of shame. Thus, while the findings for external shame are promising, they remain less conclusive.

The evidence base is considerably weaker for the interpersonal components of Gilbert’s model. Compassion to others and compassion from others were measured in only three to four studies each, with small and inconsistent effects. These studies had very small samples and were rated as weak to moderate quality. Several factors likely contribute to this limited focus: the relatively recent development of validated interpersonal compassion measures, practical constraints such as questionnaire length, the prioritisation of broad clinical outcomes over specific mechanisms, and the historical emphasis on self-directed compassion in CFT research. This lack of evidence presents challenges for clinicians, who currently have limited guidance on how CFT targets these relational aspects. Consequently, clinical choices about whether CFT should be delivered individually or as a group, due to the interpersonal nature of compassion, may be hindered. Gaining further clarity on interpersonal compassion will support the development of tailored approaches for clients in whom interpersonal dynamics are central to distress and recovery.

Internal shame has also received limited research attention. Only a handful of studies have examined this construct, typically with small samples and weak methodological designs. Although earlier reviews called for further investigation into CFT’s impact on shame ([15]), internal shame remains underexplored. Even when subscale data were available, many studies reported only global shame scores. Additionally, concerns about participant burden and the acceptability of assessing shame, given its distressing and potentially shame-inducing nature, may deter inclusion in clinical research. This narrow focus limits clinicians’ ability to understand how CFT influences internal shame. Such uncertainty hinders optimisation of treatment approaches, particularly for populations in whom internal shame plays a central role in maintaining psychological distress. Without clearer evidence, it is also difficult to determine whether CFT affects internal and external shame through similar or distinct mechanisms.

### 4.2. Appraisal of Included Evidence

Several methodological limitations affect the interpretation and generalisability of findings across the evidence base. Heterogeneity in intervention protocols represents a significant challenge for synthesis and clinical implementation. CFT formats ranged from brief six-session interventions to extended 20-session models, differing in delivery mode, session length, and therapeutic content. Most were delivered in group formats (typically six to eight participants) over 12 weekly sessions. Group settings may offer therapeutic benefits such as social support and opportunities to practise interpersonal compassion ([55]; [5]), but comparative studies of group versus individual delivery are lacking. This limits our understanding of optimal implementation strategies and clinical scalability.

Risk of bias was consistently high across the included studies. Common methodological concerns included a lack of participant masking, high attrition, selection bias in recruitment procedures, and potential co-intervention in some studies. These issues compromise the internal validity of the findings and reduce confidence in the reported effects. Many studies have also used single-arm designs without control groups, limiting the ability to attribute outcomes specifically to CFT.

Another critical limitation is the lack of follow-up data. Only six studies included follow-up data beyond immediate post-intervention timepoints, making it difficult to evaluate the long-term durability of CFT effects. This represents a significant gap, given that sustained therapeutic benefit is a key consideration for clinical implementation and cost-effectiveness analyses.

Demographic homogeneity remains a significant limitation affecting the generalisability of the evidence base. The pronounced gender imbalance and limited inclusion of non-Western populations and underrepresented clinical groups may obscure culturally specific experiences of compassion, shame, and self-criticism. This lack of diversity restricts the applicability of findings to broader clinical populations and highlights the need for future research to prioritise more representative sampling to better capture variations across gender, culture, and clinical presentation.

### 4.3. Appraisal of Review Processes

The review followed a rigorous and systematic methodology to enhance reliability and validity. It focused uniquely on outcomes integral to CFT theory. Grey literature was included to reduce publication bias; however, only one unpublished dissertation was identified ([61]), which was later published. The published, peer-reviewed version was included to minimise bias. The search was limited to English-language studies due to practical constraints. The English-language limitation may have introduced bias, affecting the representativeness of the included evidence ([17]).

While the review applied rigorous inclusion criteria to ensure relevance to clinical practice, these criteria may have inadvertently limited the representation of diverse populations. For example, a study examining CFT for transgender and gender non-conforming individuals was excluded due to the absence of formal clinical classification measures, despite addressing substantial psychological distress ([79]). Studies involving individuals with substance use or misuse as a primary diagnosis were also excluded. Although such populations could reasonably be included in future reviews, substance use fell outside the defined scope of this review. These exclusions limit the generalisability of the findings and prohibit important perspectives. This reflects a broader methodological tension in systematic reviews: balancing clear inclusion standards with the need for comprehensive and inclusive representation ([69]).

To minimise selection bias, an anonymous, independent reviewer assessed a randomly selected subset of records (10% for screening; 20% for quality assessment), achieving perfect agreement in both cases (κ = 1.00). The quality assessment used the validated EPHPP tool ([82]), which has demonstrated content validity and test–retest reliability ([2]; [16]). However, as the primary investigator conducted most of the review, there remains potential for confirmation bias. This review followed PRISMA guidelines, was pre-registered with PROSPERO, and adhered to SWiM guidance for methodological clarity and transparency.

### 4.4. Implications

#### 4.4.1. Practice

CFT interventions demonstrate the strongest potential for improving self-compassion and reducing self-criticism in clinical populations. Significant improvements were observed in relatively short group sessions, highlighting the potential for efficient clinical implementation. However, further research is needed to understand the efficacy of CFT in individual formats. Clinicians seeking to address self-criticism, shame, and compassion-related difficulties may find CFT techniques valuable. Additionally, therapists may wish to use the outcome measures included in the reviewed studies to evaluate intervention efficacy.

#### 4.4.2. Research

Future research should address several gaps. First, recruitment should prioritise underrepresented populations to enhance generalisability and ensure equitable access to compassion-based interventions. This includes participants across all ages, gender identities and sexual orientations, diverse racial/ethnic backgrounds and socioeconomic levels, people of all abilities and neurotypes, those with substance use disorders, and non-Western samples. Second, standardised measures should be used across studies to facilitate comparison and synthesis. For example, the CEAS ([34]) for compassion flows and the EISS ([19]) for shame states. Third, more rigorous research designs are needed to assess long-term impacts and compare group versus individual delivery formats. Fourth, identifying the specific CFT techniques that drive change could refine therapeutic delivery. Finally, standardised measures of treatment fidelity are needed to assess therapist adherence and participant engagement with compassion practices between sessions. Currently, the field lacks validated tools to evaluate whether CFT is being delivered as intended or to measure the extent to which participants incorporate compassion exercises into their daily lives. Understanding these factors would help clarify which components of CFT contribute most significantly to outcomes and whether the degree of participant engagement with compassion practices mediates therapeutic benefits.

#### 4.4.3. Policy

CFT cannot yet be confidently included in best practice guidelines without higher-quality evidence addressing current limitations. Continued funding for methodologically robust CFT research is crucial, as is increased awareness of CFT training opportunities. Policymakers should take a cautiously optimistic approach while supporting ongoing scientific investigation into this promising therapeutic approach.

## 5. Conclusions

This systematic review found that CFT effectively improves self-compassion and reduces self-criticism in clinical populations, with most studies reporting medium to large effect sizes. It provides promising evidence that CFT may reduce external shame, with consistent findings across studies measuring this outcome. However, internal shame remains understudied despite its theoretical importance. The review highlights significant measurement gaps in the interpersonal compassion flows (compassion to others and receiving compassion from others) with mixed findings from the few studies examining these outcomes. These patterns suggest that while CFT shows robust effects on self-directed processes, its impact on relational aspects of compassion requires further investigation. Future research should address methodological limitations, compare delivery formats, and use consistent, theory-driven outcome measures. Clinicians should consider CFT for targeting shame and self-criticism, while policymakers should recognise CFT’s emerging potential within clinical populations.

## Figures and Tables

**Figure 1 behavsci-15-01031-f001:**
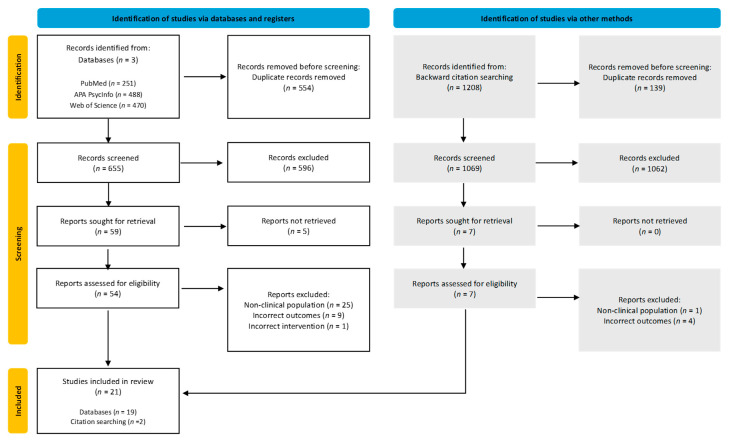
PRISMA flow diagram of the search strategy.

**Figure 2 behavsci-15-01031-f002:**
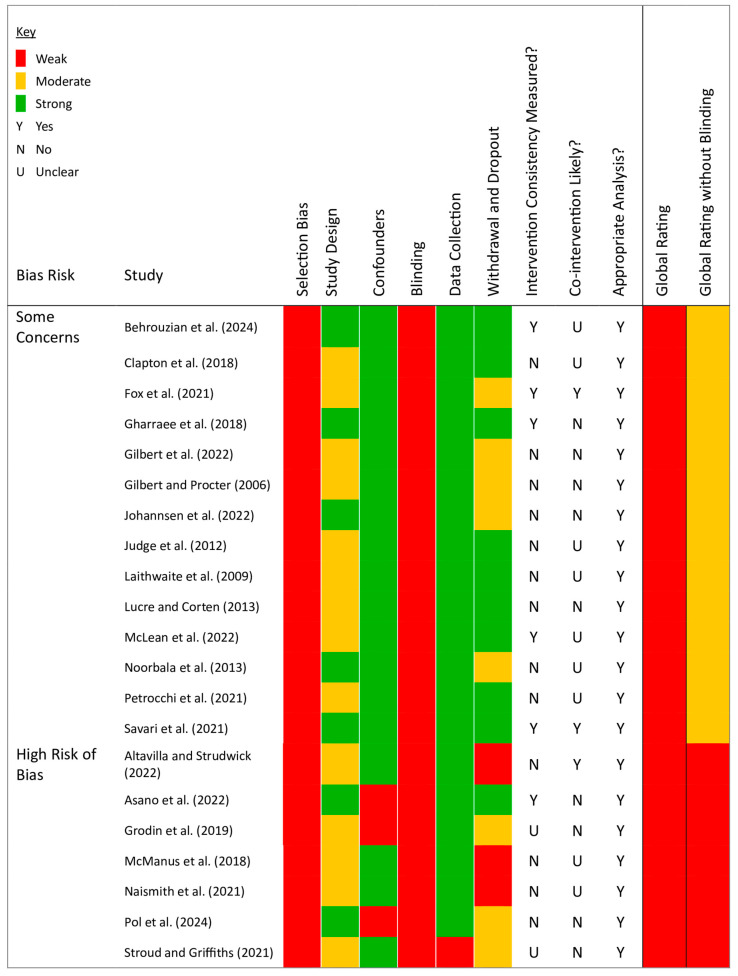
Quality appraisal within studies using the EPHPP tool. The figure includes studies by [1] ([1]), [4] ([4]), [8] ([8]), [12] ([12]), [22] ([22]), [23] ([23]), [33] ([33]), [39] ([39]), [43] ([43]), [47] ([47]), [48] ([48]), [53] ([53]), [60] ([60]), [61] ([61]), [62] ([62]), [65] ([65]), [67] ([67]), [70] ([70]), [71] ([71]), [78] ([78]), and [81] ([81]).

**Figure 3 behavsci-15-01031-f003:**
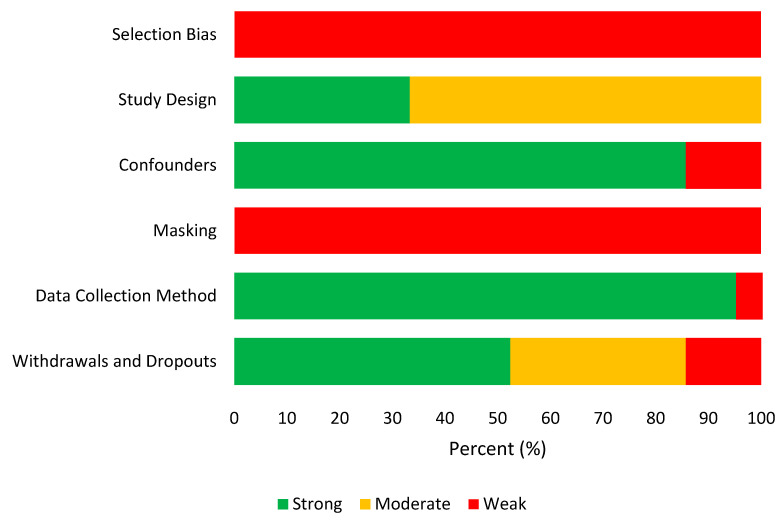
Quality appraisal across studies using the EPHPP tool.

**Figure 4 behavsci-15-01031-f004:**
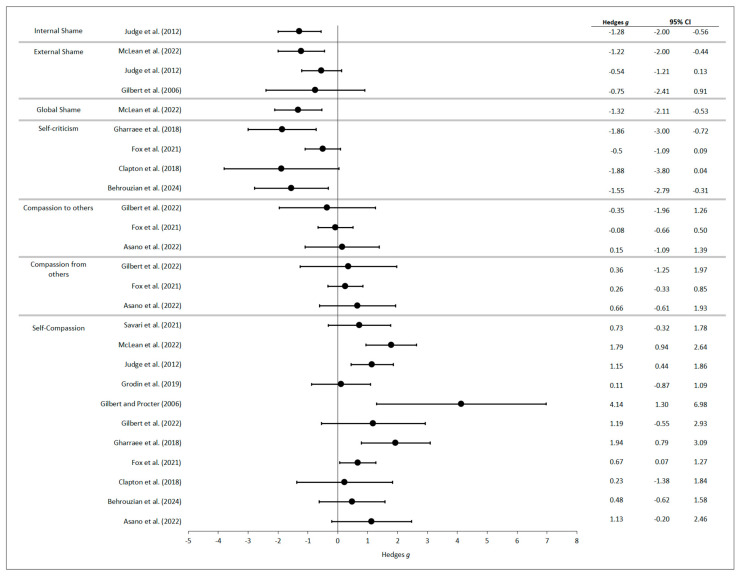
Forest plot summarising standardised effect sizes (Hedges’ *g*) for CFT outcomes across the included studies. The figure includes studies by [4] ([4]), [8] ([8]), [12] ([12]), [22] ([22]), [23] ([23]), [33] ([33]), [39] ([39]), [43] ([43]), [48] ([48]), [61] ([61]), and [78] ([78]). Notes: Effect sizes (Hedges’ *g*) are grouped by outcome: self-compassion, compassion from others, compassion to others, self-criticism, global shame, external shame, and internal shame. Error bars represent 95% confidence intervals. Positive values indicate an increase. Negative values indicate a decrease. Values reflect pre- to post-intervention changes. Only studies for which Hedges’ *g* could be calculated based on available data are included. This figure provides a descriptive visual summary and is not based on meta-analysis.

**Table 1 behavsci-15-01031-t001:** Inclusion and exclusion criteria within the PICOS framework.

PICOS	Inclusion	Exclusion
Population	Adults aged 18 and above. Pre-existing mental health condition and/or meeting the criteria of a mental health condition in accordance with a diagnostic tool (for example. DSM-5, ICD-10) indicated by a senior clinician, psychiatrist, and/or medical records. Recruited from clinical mental health settings (for example, services ranging from maximum-secure to inpatient and community settings). Scoring above a cut-off point on a relevant screening measure (for example, the General Anxiety Disorder-7 measure for anxiety).	Non-clinical populations. This includes Substance misuse disorders treated within specialist substance misuse services. Studies without reported mental health diagnoses and with screening measure means below 1 standard deviation of clinical cut-offs.
Intervention	Group or individual CFT derived from [27] ([27], [29], [32]), including psychoeducation (for example, the tricky brain, three regulatory systems, fear of compassion, and the role of shame and self-criticism) and exercises (for example, soothing rhythm breathing, mindful attention, and compassionate letter writing and/or imagery).	Any other interventions that do not constitute as CFT or do not cover the key components of the model, including other compassion-based interventions (for example, Neff’s mindful self-compassion, mindfulness-based cognitive therapy, and compassion cultivation training).
Comparison	Any alternative psychological intervention or control group (e.g., waitlist control or treatment-as-usual). Studies with no comparator.	
Outcome	Pre- and post-intervention outcomes for: Self-compassion, compassion to and from others; Self-criticism/self-reassurance; Internal and/or external shame.	Studies that do not measure CFT-based outcomes. Studies that measure physiological outcomes only, such as heart rate variability.
Study Design	Cohort studies. Non-randomised designs. Controlled clinical trials. Randomised controlled trials, feasibility studies, and pilot studies. Single case experimental designs.	Absence of pre–post statistical analysis or reliable change indices. Case studies (*n* = 1). Cross-sectional studies. Studies using only qualitative methodologies.

**Table 2 behavsci-15-01031-t002:** Standardised mean difference classifications.

Effectiveness	Very Small	Small	Medium	Large
Cohen’s *d* ([13])	<0.20	0.20	0.50	≥0.80
Hedges’ *g* ([13])		0.20	0.50	≥0.80
Rosenthal’s *r* ([76])		0.10	0.30	≥0.50

**Table 3 behavsci-15-01031-t003:** Characteristics of included studies.

First Author	Clinical Group		Sample Size		Sample Details	Intervention (Reference)	Comparator/Control Group	Individual or Group Delivery	Session Number, Duration in Hours (h), and Frequency	Facilitator(s)
Location	(Diagnostic Tool)	Study Diagnostic Interview (Yes/No); Diagnosis Date; Diagnostic Evidence	Included Sample	Retention Rate (%)	Age (*M*, *SD*, Range) Gender (% Female, % Male)					Facilitator Training
[1] ([1]) UK	Depression, anxiety, bipolar disorder, PTSD, and psychosis, recruited from an NHS secondary mental health setting. (NR cut-offs on CORE-OM, DASS-21)	No; Unspecified; Medical records.	23	22/23 (95.65%)	*NR*, *NR*, 32–89 NR, NR	CFT ([27], [28]; [57]; [88])	N/A	Group (Four cohorts with 6–10 people)	20, 2.5 h, weekly	Two clinical psychologists. Clinical work with CFT.
[4] ([4]) Japan	Treatment- resistant depression, recruited from the community. (≥20 on the BDI)	Yes; Unspecified; The Japanese M.I.N.I	17 CFT: *n* = 10 TAU: *n* = 7	Total: 17/18 (94.44%) CFT: 9/10 (90.00%) TAU: 7/7 (100.00%)	CFT: *39.80*, *11.22*, 24–56 80.00%, 20.00% TAU: *40.00*, *11.46*, 26–55 85.00%, 15.00%	CFT ([48])	TAU	Group	12, 1.5 h, weekly	One clinical psychologist, one industrial counsellor. 1/2 facilitators attended 3-day CFT workshop. Both CBT trained and supervised by PG.
[8] ([8]) Iran	Bipolar disorder, recruited from a psychiatric clinic.	Yes; Unspecified; Bipolar disorder diagnosed by an expert clinician using the SCID-I	26 CFT: *n* = 13 TAU: *n* = 13	Total: 26/31 (86.66%) CFT: 13/16 (81.25%) TAU: 13/15 (86.66%)	CFT: *29.15, 6.44*, NR 84.62%, 15.38% TAU: *28.46, 5.86*, NR 76.92%, 23.08%	CFT ([28], [29]; [23])	TAU	Individual	10, NR, weekly	NR NR
[12] ([12]) UK	Depression and anxiety, recruited from an NHS Intellectual Disabilities service. (NR diagnostic tools)	No; Unspecified; Medical records.	7	6/7 85.71%	*38.5, 15.6*, NR 66.70%, 33.30%	CFT ([28], [29]; [37]; [39]; [46])	N/A	Group (Two cohorts with three people)	6, 1.5 h, weekly	Two clinical psychologists, one trainee clinical psychologist. Lead facilitators with three years of CFT experience and supervision by PG.
[22] ([22]) USA	Distress at a clinical level, recruited from university counselling services. (≥64 on the OQ-45)	No; Unspecified; Unspecified.	91	45/91 49.45%	*22.70, NR,* 18–29 Among 75 participants who completed pre-treatment measures: 73.50%, 26.50%.	CFT ([27], [31]; [35]).	N/A	Group (Eight cohorts of 7–14 people) One individual consolidation session every three weeks	12, 2 h, weekly	Five clinical psychologists. 3/5 facilitators trained and supervised by PG.
[23] ([23]) Iran	Social anxiety disorder, recruited from clinical settings. (LSAS pre-treatment mean 74.1 and 73.1 for CFT and waitlist, respectively)	Yes; 2017–2018; SCID-I/CV conducted by two assessors (psychiatrist and clinical psychologist).	34 CFT: *n* = 17 Waitlist: *n* = 17	Total: 32/34 94.12% CFT: 17/17 100.00% Waitlist: 15/17 88.24%	CFT: *22.7, 4.58*, NR 47.10%, 52.90% Waitlist: *22.00, 4.39*, NR 47.10%, 52.90%	CFT for social anxiety disorder ([9], social anxiety protocol informed by [28], [29]; [39]).	Waitlist	Individual	12, 1 h, weekly	One clinical psychologist. Protocol adherence was assessed by an independent and anonymous clinical psychologist for 20% of sessions.
[33] ([33]) UK	Bipolar disorder, recruited from a specialist NHS Bipolar Service.	No; Unspecified; Medical records.	13	6/13 46.15%	*NR, NR*, 31–60 40.00%, 60.00%	CFT ([25], [28])	N/A	Group	25, NR, ~weekly	Two clinical psychologists. Trained and supervised by PG.
[39] ([39]) UK	Personality disorder, recruited from an NHS Day Centre. (HADS anxiety and depression in clinical range)	No; Unspecified; Psychiatric diagnosis diagnosed a psychiatrist.	9	6/9 66.67%	*45.20, 5.54*, 39–51 66.60%, 33.30%	CFT ([37])	N/A	Group	12, 2 h, weekly	Facilitated by PG and a cognitive therapist.
[43] ([43]) USA	PTSD, recruited from a specialist outpatient clinic. (≥31 on PCL-5)	No; Unspecified; Psychiatric diagnosis.	22	16/22 72.73%	*52.60, 12.90*, NR 4.00%, 96.00%	CFT ([51])	N/A	Group	12, NR, NR	Four clinical psychologists (two facilitators in each group). Trained and supervised by RK.
[47] ([47]) Denmark	Prolonged grief disorder, recruited from a bereavement study. (≥25 on the PG-13)	No; Unspecified; Diagnostic tool.	Total: *n* = 82 CFT: *n* = 42 Waitlist: *n* = 40	Total: 61/82 74.39% CFT: 26/42 61.90% Waitlist: 35/42 83.33%	Total: *60.50, 13.40*, 23–83 67.10%, 32.90% CFT: *61.40, 13.60*, NR 71.40%, 28.60% Waitlist: 59.50, 13.80, NR 62.5%, 37.5%	CFT adapted for prolonged grief. ([27], [29])	Waitlist	Group	8, 2.25 h, weekly	Three clinical psychologists. Trained and supervised by PG.
[48] ([48]) UK	Transdiagnostic, recruited from NHS Community Mental Health Teams. (≥17 on the BDI)	No; Unspecified; Medical records.	42	36/42 completed sessions: 85.71%. 27/42 completed measures: 64.29%.	*40.90, 8.80*, 22–56 59.30%, 40.70%	CFT ([27])	N/A	Group (Seven cohorts of ~five people)	12–14, 2.25 h, weekly	NR Trained and supervised by PG.
[53] ([53]) UK	Schizophrenia, Schizo-affective or bipolar affective disorder, recruited from a maximum- secure hospital. (NR)	No; Unspecified; Unspecified.	19	18/19 94.74%	*36.90, 9.10*, NR 0.00%, 100.00%	CFT ([25])	N/A	Group (Three cohorts)	20, NR, twice weekly	Two clinical psychologists, one advanced practitioner, one trainee clinical psychologist, and two assistant psychologists. NR
[60] ([60]) UK	Personality disorder, recruited from a specialist NHS Personality Disorder Service.	Yes; 2010; International PD examination completed by an NHS Cognitive Behavioural Psychotherapist.	10	8/10 80.00%	*NR, NR*, 18–54 77.70%, 22.20%	CFT ([25])	N/A	Group	16, NR, NR	Two psychotherapists. Trained and supervised by PG.
[61] ([61]) Australia	Transdiagnostic, recruited from specialist Sexual Assault services. (≥31 on PCL-5 or ≥10 and ≥8 on DASS-21 on depression and anxiety subscales, respectively).	No; No; Unspecified.	36	Pre to post-CFT: 30/36 83.33% Pre to follow-up: 25/36 69.44%	*41.10, 13.30*, 18–65 100.00%, 0.00%	CFT ([57]; [27], [30])	N/A	Group (Four cohorts of 7–11 people)	12, 2 h, weekly	One psychologist, one counsellor. Psychologist had 20 years’ clinical experience working with survivors of interpersonal trauma; trained and supervised by PG.
[62] ([62]) UK	Transdiagnostic, recruited from NHS Community Mental Health Teams. NR	No; Unspecified; Medical records.	27	13/27 48.15%	43, NR, 23–67 46.20%, 53.80%	CFT ([10]; [28]; [88])	N/A	Group (Four cohorts) One individual session (week 5)	16, 2 h, weekly	Two facilitators per group: either clinical psychologists or trainee clinical psychologists. NR
[65] ([65]) Colombia	Transdiagnostic, recruited from non-governmental organisations for low-income women. (either ≥34 on IES-R, or ≥10 on PHQ-9, or ≥8 on GAD-7).	No; Unspecified; Unspecified.	41	10/41 24.39%	*37.80, 11.70*, 19–52 100.00%, 0.00%	CFT ([29])	N/A	Group	5, 2.25 h, weekly	One clinical psychologist, one marriage and family therapist. NR
[67] ([67]) Iran	Depression, recruited from a psychiatric clinic. (≥20 on BDI)	Yes; Unspecified; Psychiatrist diagnosed major depressive disorder according to DSM-IV criteria.	Total: *n* = 22 CFT: *n* = 11 Waitlist: *n* = 11	Total: 19/22 86.36% CFT: 9/11 81.82% Waitlist: 10/11 90.91%	*28.20, NR*, 20–40 100.00%, 0.00%	CFT ([26])	Waitlist	Group	12, 2 h, twice weekly	NR NR
[70] ([70]) Italy	Obsessive compulsive disorder, recruited from an anxiety and mood disorders unit. (≥14 on Y-BOCS)	Yes; Unspecified; OCD diagnosed by an expert clinician using the SCID-I	8	8/8 100.00%	*NR, NR*, 34–41 50.00%, 50.00%	CFT ([35])	N/A	Group	8, 2 h, weekly	Two psychotherapists. One with 8 years of training with PG, one with five years of training in CFT.
[71] ([71]) The Netherlands	Personality disorder, recruited from a day-hospital. (≥14.93 on BPDSI-IV)	No; Unspecified; Unspecified	12	9/12 75.00%	*39.30, 44.30*, NR 100.00%, 0.00%	CFT ([38]; [72])	N/A	Group	12, NR, weekly	One clinical psychologist. Basic and advanced training in CFT with PG.
[78] ([78]) Iran	Major depressive disorder, recruited from recruited from university counselling services. (≥20 on BDI)	Yes; NR; SCID-I	Total: 30 CFT: 15 Waitlist: 15	Total: 30/30 100.00% CFT: 15/15 100.00% Waitlist: 15/15 100.00%	*24.30, 2.16*, 21–29 100.00%, 0.00%	CFT ([28]; [35])	N/A	Group	8, 1.5 h, twice weekly for 4 weeks	Senior author. Attended CFT training with PG.
[81] ([81]) UK	Transdiagnostic, recruited from an inpatient mental health setting. (≥10 on CORE-OM)	No; Unspecified; Unspecified.	Total: *n* = 32 CFT: *n* = 19 TAU: *n* = 13	NR	*NR, NR*, 18–60 59.40%, 40.60%	TAU & CFT (Gilbert, year unspecified)	TAU	Group (open format)	6, 1 h, daily (Repeated cyclically every week over 4-months)	NR NR

Notes: NR = Not Reported; PG = Paul Gilbert; RK = Russel Kolts; CORE-OM = Clinical Outcomes in Routine Evaluation—Outcome Measure ([18]); BDI = Beck’s Depression Inventory ([6]); Japanese M.I.N.I = Mini-international Neuropsychiatric Interview ([68]); DASS-21 = Depression Anxiety and Stress Scale ([59]); OQ-45 = Outcome Questionnaire-45 ([54]); SCID-I/CV = Structured Clinical Interview for DSM-IV Axis I Disorders, Clinical Version ([21]); PG-13 = Prolonged Grief-13 ([73]); PCL-5 = PTSD Checklist for DSM-5 ([86]); IES-R = Impact of Event Scale-Revised ([87]); PHQ-9 = Patient Health Questionnaire ([52]); GAD-7 = Generalised Anxiety Disorder Assessment ([80]); Y-BOCS = Yale-Brown Obsessive-Compulsive Scale ([41]); BPDSI-IV = The Borderline Personality Disorder Severity Index-IV ([3]); SCID-I = Structured Clinical Interview for Diagnosis ([20]).

**Table 4 behavsci-15-01031-t004:** Results across studies.

First Author	Design	Outcome Measures	Timepoints	Main Outcome(s)
Location		(Measuring Tools)		(Reported *p,* ES (*d*), and Calculated *g*)
[1] ([1]) UK	Mixed methods (Cohort: One group pre + post, and qualitative interviews)	Self-compassion (SCS)	Pre-intervention, Session 20, 3-month follow-up	^a^ Significant increase in self-compassion (SCS) post-intervention (*p* = 0.008)
[4] ([4]) Japan	RCT (CFT vs. TAU)	Self-compassion (CEAS) Compassion from others (CEAS) Compassion to others (CEAS)	Pre-intervention, post-intervention	Change in self-compassion (CEAS) post-intervention (*d* = 1.08 ^L^, *g* = 1.13 ^L^) compared to TAU. Change in compassion from others (CEAS) post-intervention (*d* = 0.61 ^M^, *g* = 0.66 ^M^) compared to TAU. Change in compassion to others (CEAS) post-intervention (*d* = 0.14 ^VS^, *g* = 0.15 ^VS^) compared to TAU.
[8] ([8]) Iran	RCT (CFT vs. TAU)	Self-compassion (SCS-SF) Self-criticism (SCRS)	Pre-intervention, post-intervention, and 2-month follow-up	No significant change in self-compassion (SCS-SF) post-intervention compared to TAU (*p* = 0.49, *d* = 0.50 ^M^, *g* = 0.48 ^S^). Significant decrease in self-criticism (SCRS) post-intervention compared to TAU (*p* < 0.001, *d* = −1.60 ^L^, *g* = −1.55 ^L^).
[12] ([12]) UK	Mixed methods (Cohort: One group pre + post, and qualitative interviews)	Self-compassion (SCS-SF) Self-criticism (SCS-SF)	Pre-intervention, 2–4 weeks post-intervention	No significant change in self-compassion (SCS-SF) post-intervention (*p* = 0.674, *d* = 0.25 ^S^, *g* = 0.23 ^S^). Significant decrease in self-criticism (SCS-SF) post-intervention (*p* = 0.027, *d* = −2.04 ^L^, *g* = −1.88 ^L^).
[22] ([22]) USA	Cohort: One group pre + post	Self-compassion (CEAS) Compassion from others (CEAS) Compassion to others (CEAS) Self-criticism (Inadequate Self, Hated Self & Reassured Self; FSCRS) Self-criticism (DEQ)	Pre-intervention, session 6, post-intervention (session 12+)	Significant increase in self-compassion (CEAS) post-intervention (*p* < 0.001, *d* = 0.75 ^M^, *g* = 0.67 ^M^). Significant increase in compassion from others (CEAS) post-intervention (*p* < 0.001, *d* = 0.26 ^S^, *g* = 0.26 ^S^). No significant change in compassion to others (CEAS) post-intervention (*p* = 0.19, *d* = −0.08 ^VS^, *g* = −0.08 ^VS^). Significant decrease in inadequate self and hated self (FSCRS) post-intervention (*p* < 0.001, *d* = −0.71 ^M^, *g* = −0.70 ^M^, and *p* < 0.001, *d* = −0.34 ^S^, *g* = −0.33 ^S^, respectively). Significant increase in reassured self (FSCRS) post-intervention (*p* < 0.001, *d* = 0.40 ^S^, *g* = 0.40 ^S^). Significant decrease in self-criticism (DEQ) post-intervention (*p* < 0.001, *d* = −0.50 ^M^, *g* = −0.50 ^M^).
[23] ([23]) Iran	RCT (CFT vs. Waitlist)	Self-compassion (SCS) Self-criticism (LOSC)	Pre-intervention, post-intervention, and 2-month follow-up	Significant increase in self-compassion (SSC) compared to waitlist control at post-intervention (*p* < 0.001, *d* = 1.99 ^L^, *g* = 1.94 ^L^), which was maintained at follow-up (*p* = 0.39, *d* = −0.37 ^S^, *g* = −0.36 ^S^). Significant decrease in self-criticism (LOSC) compared to waitlist control at post-intervention (*p* < 0.001, *d* = −1.91 ^L^, *g* = −1.86 ^L^), which further increased at follow-up (*p* < 0.001, *d* = 0.26 ^S^, *g* = 0.25 ^S^).
[33] ([33]) UK	Mixed methods (Cohort: One group pre + post, and qualitative interviews)	Self-compassion (CEAS) Compassion from others (CEAS) Compassion to others (CEAS) Self-criticism (Inadequate Self, Hated Self & Reassured Self; FSCRS)	Pre-intervention, 12 weeks, 32 weeks, and post-intervention (45 weeks)	An overall improvement in self-compassion and compassion from others (CEAS) were reported from baseline to 45 weeks (6 out of 6 participants improved, *d* = 1.29 ^L^, *g* = 1.19 ^L^, and 4 out of 6 participants improved, *d* = 0.39 ^S^, *g* = 0.36 ^S^, respectively). No overall improvement in compassion to others (CEAS) was reported from baseline to 45 weeks (3 out of 6 participants improved, *d* = −0.38 ^S^, *g* = −0.35 ^S^). An overall improvement in inadequate self, hated self, and reassured self (FSCRS) were reported from baseline to 45 weeks (6 out of 6 participants improved, *d* = −1.21 ^L^, *g* = −1.11 ^L^, 2 out of 6 participants improved, *d* = −0.21 ^S^, *g* = −0.19 ^VS^, and 5 out of 6 improved, *d =* 0.61 ^M^, *g* = 0.56 ^M^, respectively).
[39] ([39]) UK	Cohort: One group pre + post	Self-compassion (Weekly Interval Contingent Diary Measuring Self-attacking and Self-soothing) Self-criticism (Inadequate Self, Hated Self & Reassured Self; FSCRS) External shame (OAS)	Pre-intervention (beginning of week 1), post-intervention (end of week 12)	Significant increase in self-compassion (Diary) post-intervention (*p* = 0.030, *d* = 4.49 ^L^, *g* = 4.14 ^L^). No significant change in inadequate self (FSCRS) post-intervention (*p* = 0.070, *d* = −2.73 ^L^, *g* = −2.52 ^L^). Significant decrease in hated self (FSCRS) post-intervention (*p* = 0.030, *d* = −2.04 ^L^, *g* = −1.88 ^L^). Significant increase in reassured self (FSCRS) post-intervention (*p* = 0.030, *d* = 1.86 ^L^, *g* = 1.71 ^L^). Significant decrease in external shame (OAS) post-intervention (*p* = 0.030, *d* = −0.82 ^L^, *g* = −0.75 ^M^).
[43] ([43]) USA	Cohort: One group pre + post (pilot study)	Self-compassion (SCS)	Pre- and post-intervention	No significant change in self-compassion (SCS) post-intervention (*p* = 0.34, *d* = 0.17 ^VS^, *g* = 0.11 ^VS^).
[47] ([47]) Denmark	RCT (CFT vs. Waitlist)	Self-criticism (Inadequate Self, Hated Self & Reassured Self; FSCRS)	Pre-intervention, post-intervention, 3-month and 6-month follow-up	No significant change in inadequate self and hated self (FSCRS) compared to waitlist post-intervention (*p* = 0.24, *d* = −0.44 ^S^, *g* = −0.43 ^S^ and *p* = 0.15, *d* = −0.08 ^VS^, *g* = −0.08 ^VS^, respectively). Significant increase in reassured self (FSCRS) compared to waitlist post-intervention (*p* = 0.001, *d* = 0.43 ^S^, *g* = 0.43 ^S^).
[48] ([48]) UK	Cohort: One group pre + post	Self-compassion (Weekly Interval Contingent Diary Measuring Self-attacking and Self-soothing) Self-criticism (Inadequate Self, Hated Self & Reassured Self; FSCRS) Internal shame (ISS) External shame (OAS)	Pre- and post-intervention	Significant increase in self-compassion (Diary) post-intervention (*p* < 0.001, *d* = 1.17 ^L^, *g* = 1.15 ^L^). Significant decrease in inadequate self and hated self (FSCRS) post-intervention (*p* < 0.001, *d* = −1.35 ^L^, *g* = −1.33 ^L^ and *p* < 0.001, *d* = −0.90 ^L^, *g* = −0.88 ^L^, respectively). Significant increase in reassured self (FSCRS) post-intervention (*p* < 0.001, *d* = 0.93 ^L^, *g* = 0.91 ^L^). Significant decrease in internal shame (ISS) post-intervention (*p* < 0.001, *d* = −1.30 ^L^, *g* = −1.28 ^L^). Significant decrease in external shame (OAS) post-intervention (*p* < 0.001, *d* = −0.55 ^M^, *g* = −0.54 ^M^).
[53] ([53]) UK	Cohort: One group pre + post	Self-compassion (SCS) External shame (OAS)	Pre-intervention, mid-group (5 weeks), post-intervention, 6-week follow-up	^a^ No significant change in self-compassion (SCS) post-intervention (*p* = 0.180, *r* = 0.22 ^S^). ^a^ Significant decrease in external shame (OAS) post-intervention (*p* = 0.040, *r* = 0.04 ^<S^).
[60] ([60]) UK	Mixed methods (Cohort: One group pre + post, and qualitative interviews)	Self-criticism (Inadequate Self, Hated Self & Reassured Self; FSCRS) External shame (OAS)	Pre- and post-intervention (16 weeks), 1-year follow-up	^a^ No significant change in inadequate self (FSCRS) at follow-up (*p* = 0.062). ^a^ Significant decrease in hated self (FSCRS) post-intervention, which was maintained at follow-up (*p* < 0.001). ^a^ Significant increase in reassured self (FSCRS) post-intervention, which was maintained at follow-up (*p* < 0.001). ^a^ Significant decrease in external shame (OAS) post-intervention, which further decreased at follow-up (*p* = 0.011).
[61] ([61]) Australia	Cohort: One group pre + post	Self-compassion (CEAS) Self-criticism (Inadequate Self, Hated Self & Reassured Self; FSCRS) Global shame (EISS) External shame (OAS)	Pre-intervention (2 weeks pre-group), post-intervention (12 weeks), 3-month follow-up	Significant increase in self-compassion (CEAS) post-intervention (*p* < 0.001, *d* = 1.82 ^L^, *g* = 1.79 ^L^) and at follow-up (*p* < 0.001, *d* = 1.57 ^L^, *g* = 1.55 ^L^). Significant decrease in inadequate self (FSCRS) post-intervention (*p* < 0.001, *d* = −1.59 ^L^, *g* = −1.56 ^L^) and at follow-up (*p* < 0.001, *d* = −1.73 ^L^, *g* = −1.70 ^L^). Significant decrease in hated self (FSCRS) post-intervention (*p* < 0.001, *d* = −0.98 ^L^, *g* = −0.97 ^L^) and at follow-up (*p* < 0.001, *d* = −0.94 ^L^, *g* = −0.92 ^L^). Significant increase in reassured self (FSCRS) post-intervention (*p* < 0.001, *d* = 1.20 ^L^, *g* = 1.18 ^L^) and at follow-up (*p* < 0.001, *d* = 1.11 ^L^, *g* = 1.09 ^L^). Significant decrease in global shame (EISS) post-intervention (*p* < 0.001, *d* = −1.70 ^L^, *g* = −1.67 ^L^) at follow-up (*p* < 0.001, *d* = −1.35 ^L^, *g* = −1.32 ^L^). Significant decrease in external shame (OAS) post-intervention (*p* < 0.001, *d* = −1.24 ^L^, *g* = −1.22 ^L^) and at follow-up (*p* < 0.001, *d* = −0.77 ^M^, *g* = −0.75 ^M^).
[62] ([62]) UK	Mixed methods (Cohort: One group pre + post, and qualitative interviews)	Self-compassion (SCS) Self-criticism (Inadequate Self, Hated Self & Reassured Self; FSCRS) External shame (OAS)	Pre- and post-intervention (week 16)	^a^ Significant increase in self-compassion (SCS) post-intervention (*p* = 0.010). ^a^ Significant decrease in inadequate self (FSCRS) post-intervention (*p* = 0.030). ^a^ Significant decrease in hated self (FSCRS) post-intervention (*p* = 0.020). ^a^ No significant change in reassured self (FSCRS) post-intervention (*p* = 0.070). ^a^ Significant decrease in external shame (OAS) post-intervention (*p* = 0.010).
[65] ([65]) Colombia	Cohort: One group pre + post	Self-criticism (Inadequate Self, Hated Self; FSCRS)	Pre- (5 weeks before intervention) and post-intervention, and 3-month follow-up	^b^ Inadequate self (FSCRS) reliably improved for 5 out of 10 participants post-intervention, and 5 out of 9 participants at follow-up. ^b^ Hated self (FSCRS) reliably improved for 4 out of 5 participants, and 3 out of 5 participants at follow-up (5 scored under 6 pre-intervention and were excluded).
[67] ([67]) Iran	Case controlled clinical trial (CFT vs. Waitlist)	Self-criticism (Comparative and Internalised; LOSC)	Pre-intervention, post-intervention, and 2-month follow-up	^a^ No significant change in comparative self-criticism (LOSC) post-intervention (*p* = 0.491) or at follow-up (*p* = 0.108). ^a^ No significant change in internalised self-criticism (LOSC) post-intervention (*p* = 0.491) or at follow-up (*p* = 0.272).
[70] ([70]) Italy	Multiple baseline design	Self-compassion (Common Humanity; SCS) Self-criticism (Inadequate Self & Reassured Self; FSCRS)	Baseline, pre-intervention, post-intervention, and 1-month follow-up	No significant change in self-compassion (Common Humanity; SCS) post-intervention (*p* = 0.11, *g* = 0.49 ^S^) and at follow-up (*p* = 0.230, *g* = 0.57 ^M^). Significant decrease in inadequate self (FSCRS) post-intervention (*p* = 0.030, *g* = −0.29 ^S^), which was not maintained at follow-up (*p* = 0.110, *g* = −0.63 ^M^). Significant increase in reassured-self (FSCRS) post-intervention (*p* = 0.040, *g* = 0.35 ^S^), which was not maintained at follow-up (*p* = 0.100, *g* = 0.32 ^S^).
[71] ([71]) The Netherlands	Multiple baseline design	Self-compassion (SCS-SF) Self-criticism (FSCRS)	Baseline, pre-intervention, post-intervention, and 6-week follow-up	^b^ Self-compassion (SCS-SF) reliably improved for 3 out of 9 participants post-intervention and 5 out of 9 at follow-up. 5 out of 9 participants and 4 out of 9 participants showed no change post-intervention and at follow-up, respectively. ^b^ Self-criticism (FSCRS) reliably improved for 6 out of 9 participants post-intervention and at follow-up. 3 out of 9 participants showed no change.
[78] ([78]) Iran	RCT (CFT vs. Waitlist)	Self-compassion (SCS-SF) Self-criticism (FSCRS; Inadequate Self, Hated Self, & Reassured Self)	Pre- and post-intervention	Significant increase in self-compassion (SCS-SF) post-intervention compared to waitlist (*p* = 0.010, *d* = 0.75 ^M^, *g* = 0.73 ^M^). No significant change in inadequate self (FSCRS) post-intervention compared to waitlist. Significant decrease in hated self (FSCRS) post-intervention compared to waitlist (*p* = 0.030, *d* = −1.00 ^L^, *g* = −0.97 ^L^). Significant increase in reassured self (FSCRS) post-intervention compared to waitlist (*p* < 0.001, *d* = 1.20 ^L^, *g* = 1.17 ^L^).
[81] ([81]) UK	Cohort analytic (Two group: CFT vs. TAU pre + post)	Self-compassion (VAS) Compassion to Others (VAS)	Pre-, every session, and post-intervention	^a^ Significant increase in self-compassion (VAS) across all sessions compared to TAU (*p* < 0.001, *d* = 0.24 ^S^). ^a^ Significant increase in compassion to others (VAS) across all sessions compared to TAU (*p* < 0.001, *d* = 0.18 ^VS^).

Notes: ^a^ *d* and/or *g* not calculated as information not available in full-text paper and no response from authors upon request for Appendix A. ^b^ *d* and *g* are not calculated as the study describes the pattern of change or uses reliable change indices only. ^VS^, ^S^, ^M^, and ^L^ indicate very small, small, medium, and large effect sizes, respectively. ^<S^ for Rosenthal’s *r* indicates effect sizes under the recommended 0.10 cut-off for a small effect. SCS = The Self-Compassion Scale; SCS-SF = The Self-Compassion Scale Short Form; CEAS = Compassionate Engagement and Action Scales; SCRS = The Self-Critical Rumination Scale; FSCRS = Forms of Self-Criticism and Self-Reassuring Scale—Inadequate Self, Hated Self, and Reassured Self Subscales; DEQ = Depressive Experiences Questionnaire 48 McGill Revision—Self-Criticism Subscale; LOSC = Level of Self-Criticism Scale; OAS = the Other as Shamer Scale; ISS = Internalised Shame Scale; EISS = External and Internal Shame Scale; VAS = Visual Analogue Scale (0–10 Likert scale for self-compassion and compassion to others developed by authors).

## Data Availability

The data that support the findings of this study are available from the corresponding author upon request. The search strategy, inclusion/exclusion criteria, and quality assessment tool are available in Appendix A.

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
