# Peer review of "The Effectiveness of Compassion Focused Therapy for the Three Flows of Compassion, Self-Criticism, and Shame in Clinical Populations: A Systematic Review"

_behavsci, 2025, doi:10.3390/bs15081031_

Round 1
Reviewer 1 Report
Comments and Suggestions for Authors
This is a thorough and thoughtful systematic review of CFT's effectiveness in changing each of the core theoretical constructs of CFT, in clinical populations (flows of compassion, self-criticism, shame). The authors examined, via 3 major databases, a wide and comprehensive array of studies of CFT in adults from 2000-2024. Of these, 21 studies were deemed to meet a rigorous set of inclusion criteria.
Overall, this is a great contribution to the psychotherapy effectiveness literature and I hope that similar meta analyses will be conducted across other treatment modalities. I do, however, have a couple of points, which, if addressed, could strengthen the readability of this manuscript.
- How were unpublished studies included if the 3 major databases search for published studies? More could be useful here.
- Relatedly, if a study is unpublished (a dissertation, or similar), might it not have been subject to the same rigorous peer review process, and might the resulting data be more questionable?
- Why exclude studies involving patients who are substance-users/misusers -this needs to be spelled out more. Currently, it is only mentioned as an item in Table 1. Given that people living with schizophrenia and other serious mental illness are included, to me, it seems a miss to exclude those with substance use/mis-use disorders.
- Were the 5 studies which were inaccessible via full text included in the final sample of 21 studies? Of the 3 for which the authors confirmed eligibility included? Could the full text not be obtained by contacting authors or similar? If these were excluded, then this is can be addressed by spelling this out more clearly. However, if 3 (or 5) of the 21 studies had not been reviewed in full text, then this gives me pause.
- Similarly, I was concerned about the quality of the study which did not report the gender of its participants. Could these data be obtained? If not, the authors could perhaps defend the inclusion of this trial despite the lack of reporting of a fairly major variable.
- Table 3 was hard to read as the formatting was wonky - since it's such a crucial table, I recommend reformatting to increase readability - several key metrics were cut off such that part was on one line, part on another, and it was hard to decipher.
- Figure 2 is v. useful - particularly calling attention to co-intervention potential.
- Overall, this is a strong paper which, if strengthened a bit more, could increase its clinical impact on the practice and review of CFT trials.
Reviewer 2 Report
Comments and Suggestions for Authors
Thank you for the opportunity to review this manuscript. I think it is excellent and will make a substantial contribution to the CFT literature.
This is such a well-done manuscript that I only have one comment (which, for me, is unusual when doing a review).
My only suggestion is to avoid using the word “blind” with respect to review and research processes. I recognize that it is a standard term, but it carries problematic connotations and can be perceived as ableist. The phrase anonymous review can be used instead.
Reviewer 3 Report
Comments and Suggestions for Authors
General Comments
This is a timely and well-conducted systematic review on the effectiveness of Compassion Focused Therapy (CFT) for self-compassion, self-criticism, and shame in clinical populations. The manuscript is clearly structured, grounded in a strong theoretical framework, and presents relevant findings for both researchers and practitioners. However, a few revisions would improve its clarity, depth, and transparency.
Title and Abstract
-
The title is informative and accurately reflects the content.
-
The abstract summarizes the main findings well but could benefit from:
-
Clarifying the strength of evidence for internal shame and interpersonal compassion.
-
Explicitly noting the limitations in terms of methodological variability and sample diversity.
-
Introduction
-
The introduction effectively contextualizes the rationale for focusing on the three flows of compassion and related constructs.
-
Consider adding a brief summary of how CFT theoretically engages with internal vs. external shame to clarify the conceptual distinction early on.
-
It might be helpful to briefly define the “three flows” for readers unfamiliar with the model.
Methods
-
The use of PRISMA guidelines and the EPHPP tool is commendable.
-
The database search is appropriate and clearly described.
-
Please specify if grey literature or unpublished studies were excluded, and why.
-
The inclusion and exclusion criteria are adequate, but a short justification for excluding non-English studies (if applicable) would enhance transparency.
-
In Table 1, adding a column to indicate whether each study was group or individual format would help highlight heterogeneity in delivery.
Results
-
The narrative synthesis is appropriate given the methodological heterogeneity.
-
The effect sizes reported are useful. Please specify how they were calculated or derived (i.e., whether reported directly or computed).
-
Consider including a figure summarizing the range of effect sizes across constructs (e.g., forest-style summary without meta-analysis).
-
Clarify why some flows of compassion were underrepresented – was it due to measurement tools or absence in intervention design?
Discussion
-
The discussion appropriately highlights the strength of CFT for self-compassion and self-criticism.
-
The limited evidence for interpersonal compassion and internal shame is well noted.
-
Please expand slightly on why these components are under-studied and how this might influence clinical implementation.
-
The recommendations for future research are sound. Including a call for studies in underrepresented clinical populations (e.g., men, non-Western samples) would enhance generalizability.
-
The discussion could better integrate the quality assessment outcomes into the interpretation of findings.
Limitations
-
Some limitations are mentioned but could be more explicit. For instance:
-
The heterogeneity in intervention protocols.
-
Risk of bias across studies.
-
The lack of follow-up measures in most included trials.
-
Language and Writing
-
The writing is clear and professional.
-
A light proofreading would help streamline some long sentences, especially in the results and discussion sections.
-
Terminology is used consistently throughout.
